# Domain Adaptation and Generalization: A Low-Complexity Approach

**Joshua Niemeijer, Jörg P. Schäfer**
German Aerospace Center (DLR)
Germany
`Joshua.Niemeijer@dlr.de`
`Joerg.Schaefer@dlr.de`

**Abstract:** Well-performing deep learning methods are essential in today's perception of robotic systems such as autonomous driving vehicles. Ongoing research is due to the real-life demands for robust deep learning models against numerous domain changes and cheap training processes to avoid costly manual-labeling efforts. These requirements are addressed by unsupervised domain adaptation methods, in particular for synthetic to real-world domain changes. Recent top-performing approaches are hybrids consisting of multiple adaptation technologies and complex training processes.

In contrast, this work proposes EasyAdap, a simple and easy-to-use unsupervised domain adaptation method achieving near state-of-the-art performance on the synthetic to real-world domain change. Our evaluation consists of a comparison to numerous top-performing methods, and it shows the competitiveness and further potential of domain adaptation and domain generalization capabilities of our method. We contribute and focus on an extensive discussion revealing possible reasons for domain generalization capabilities, which is necessary to satisfy real-life application's demands.

**Keywords:** unsupervised domain adaptation, semantic segmentation, domain generalization

## 1 Introduction

Numerous applications have become feasible due to deep learning methods for computer vision, including the environment perception of autonomous vehicles. In particular, architectures emerged to deliver high-quality results for semantic segmentation tasks, i. e., a pixel-wise classification of images. Unfortunately, it is in the nature of deep learning models to be sensitive to domain changes, e. g., domain changes from day to night, from telescope to wide-angle lenses, or from rural to urban areas. An extensive training effort is necessary to keep the high-quality on the target domain, i. e., to adapt the model from the source to the target domain. Moreover, supervised training methods for the target domains are infeasible due to the high costs of manually labeling data[1] and the virtually endless number of target domains[2].

Several approaches exist, minimizing or even completely avoiding labeling costs for the target-domain data. For example, *Active learning* approaches implement a human-in-the-loop training process, which asks a human to provide labels for a selection of samples, in order to achieve high-quality training results with minimal manual labeling effort [2]. While active learning for semantic segmentation tasks remains a huge challenge, it also still requires a certain amount of manual effort [3, 4, 5]. Unsupervised domain adaptation methods completely avoid the usage of manually labeled target-domain data. Instead, they align the distributions of source- and target-domain data in the input [6, 7, 8], feature [9, 10, 11, 12], output space [13, 14, 15], or a combination of those [16, 17, 18]. The latter is complex in design, and therefore, hard to reproduce.

---

[1] The manual pixel-wise annotation of an image takes about 90 minutes.[1]

[2] Real-life applications demand an adaptation to every domain they might encounter.

6th Conference on Robot Learning (CoRL 2022), Auckland, New Zealand.

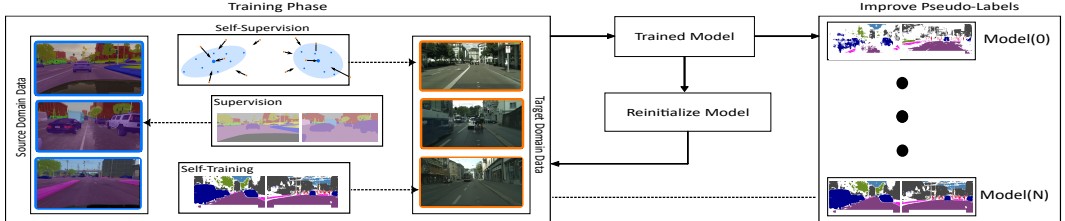

Figure 1: An overview of EasyAdapt's internal dependencies (dashed lines): The source-domain training depends on the source-domain data and labels; the self-training of the target-domain training depends on the target-domain data and its previously created pseudo-labels; the self-supervision depends on the target-domain data and the feature clusters of the source-domain data.

The usage of synthetic datasets as source domain completely abandons the need for manual labeling effort [19, 20]. However, the large domain gap between synthetic and real-world domains makes an adaptation between these domains increasingly difficult. Several aspects cause this domain gap, such as a decreased variety in synthetic data and the differences regarding geometry, texture, and lighting conditions. Due to these impediments, models trained on synthetic datasets poorly perform on real-world data. On the other hand, real-life applications demand models that are robust against a variety of domain changes. But performing domain adaptation to a large number of domains is not feasible due to the immense training effort. Desirable domain adaptation methods achieve a domain generalization. These methods train models not only to improve on the source or target domain but also on unseen domains [21, 22, 23].

This work addresses the problem of unsupervised domain adaptation from synthetic to real-world domains. We propose *EasyAdap*, an iterative extension of our work in [11], which conducts a distribution alignment in the feature- and output-space (see Figure 1 for an overview). While achieving near state-of-the-art performance on the GTA5 [19] to Cityscapes [1] domain change, our method is still simple and easy-to-tune due to its simple architecture[3]. Our feature-space distribution alignment is based on a self-supervision through a clustering loss of the target-domain features towards class-specific source-domain feature centroids. A self-training addresses the alignment of the distribution in the output space. We study the interplay between this self-supervision and self-training and show that a synergistic effect exists (see section 4.4). We discuss the reasons for this effect in Section 4.5. Overall, our work contributes EasyAdap, an easy-to-use unsupervised domain adaptation method that yields near state-of-the-art performance. We contribute an extensive study of various unsupervised domain adaptation methods regarding domain generalization.

Section 2 discusses domain adaptation methods related to our work. While Section 3 describes our method, Section 4 evaluates our approach and compares it to other domain adaptation methods w.r.t. the performance on the target domain and unseen domains. This work concludes with Section 5.

## 2   Related Work

We categorize related unsupervised domain adaptation methods for semantic segmentation tasks regarding their adaptation in the input, feature, and output space and hybrid methods thereof. The latter (hybrids) comprise methods from the different categories, making up the majority of the recent top-performing publications. While *domain adaptation* is a broad field (e. g., which includes learning new classes), we concentrate on adaptation approaches that align the distributions in the input, feature, and output space. The following subsections collect and briefly describe examples for these categories.

**Input-Space Adaptation.**   Aligning the data distribution in the input space usually resembles a style transfer from the source domain to the target domain. Since a style transfer only changes the textures while keeping the geometry of the images, the ground truth is also valid for the style-transferred images. Now, training a model on labeled style-transferred images from the source domain is close to training a model on the target domain. The most successful approaches for style

---

[3]URL: https://github.com/JNiemeijer/EasyAdap.git

transfer implement Cycle-GANs [6, 7], Fourier transformation [8, 24] or global image operators [16].

**Feature-Space Adaptation.** The adaptation in the feature space of a deep learning model usually relies on adversarial training or self-supervision and aims at distribution alignment. When the pre-logit feature distributions of the source and target domain match, the classifier in the last layer classifies the source- and target-domain images in the same way. Adversarial training is based on a feature extractor and a domain discriminator network, classifying the feature space into source and target domains. The optimization goal is to generate a feature space that is not discriminable into the source and target domain while containing relevant representations for the semantic segmentation [9, 25]. A recent approach implements a contrastive loss[4] on class-wise features [10] attracting features of the same class while repulsing features of different classes independent of their domain. The approaches presented in [12] and our approach [11] apply semantic self-supervision to align the distribution in the feature space by attracting the target-domain features in the pre-logit feature space towards class-specific source-domain centroids. In contrary to the contrastive self-supervision presented in [10] the attraction mechanism used in [11, 12] does not rely on pseudo-labels for the target domain. Our work advances the approach of [11].

**Output-Space Adaptation.** Distribution alignment approaches in the output space either implement an adversarial training or a self-training. Adversarial approaches consider the structure of the output space, i. e., they discriminate images between the source and target domain based on the (spatial) distribution of the output of the segmentation model [13, 14]. Aiming at an easily reproducible approach, we avoid adversarial approaches in our work. Most of the current top-performing methods implement a self-training on the target domain, which is a training of the model on self-inferred labels (*pseudo-labels*). The effect of self-training depends on the quality of the pseudo-labels, i. e., these methods require strategies for the mitigation of false pseudo-labels. Common approaches ignore low-confident pseudo-labels using a fixed confidence threshold [27, 28]. Some approaches filter low-confident pseudo-labels dynamically [15, 17, 29].

**Hybrid Methods.** Most of the top performing approaches are *hybrid methods* combining adaptions methods working in the input, feature, and output space, notably [16, 17, 18]. The approach presented in [18] performs a distribution alignment in the feature (clustering) and output space (self-training). The approach in [16] even combines adaptation methods in the input, feature, and output space. The currently best performing approach presented in [17] applies methods for distribution alignment in feature and output space and additionally performs three structurally different adaptation stages. Compared to these currently top-performing methods, our approach comes with a much lower complexity regarding design and implementation effort.

## 3   Method

We introduce a nested iterative training approach for domain adaptation based on repeated application of self-training and semantic clustering (see Figure 1). With this, we advance our approach in [11] from a two stage model to an iterative model. We initialize the process with a training on the source domain (see Section 3.1). The inner loop of the training process establishes a training on the source and target domain, stopping early after a short number of epochs. The training on the target domain comprises a semantic clustering and a self-training which is implemented as described in [11]. Upon finishing such an inner iteration, the outer loop of the training process creates new pseudo-labels and repeats the process. Section 3.2 describes the algorithm for the training process.

### 3.1   Building Blocks

**Source Training**   Self-training approaches require models with a sufficient understanding of the data to provide valuable pseudo-labels. To that, our approach initializes the model with a sophisticated supervised training method on the source domain, including a rich data augmentation and a class-uniform sampling strategy. Zhu et al. [30] showed significant improvements in supervised training when applying a class-uniform sampling strategy. We adapt their class-uniform sampling in

---

[4]Cf. [26] for an overview of contrastive learning.

the following way. We first gather a list of objects from the dataset, i.e., a list of polygons[5], each of which has a class attribute and a centroid. We sample from this list regarding uniformly distributed class attributes during the training. Eventually, each training batch combines randomly chosen crops and crops that are centered around these polygons. A parameter $0 \leq \gamma \leq 1$ controls the ratio of random samples and class-uniformly selected samples. We pair this uniform sampling with a strong data augmentation consisting of Gaussian blurring, color jittering, and random scaling[6].

**Semantic Clustering**    When training on the source- and target-domain data, our approach for clustering the feature space is inspired by the image-classification method presented in [12] and advances our work in [11][7]. We compute the class centroids on the source-domain data as running averages of class-specific feature representations in the pre-logit feature space. We then compute a similarity matrix between these class centroids and the feature representations of the target-domain data. After weighing each entry of the similarity matrix with a constant that scales with the certainty of the classification on the target-domain data, our loss function computes the entropy of the weighted similarity matrix. This process implements a clustering[8] algorithm that executes through the backpropagation. For details, please refer to the description in [11] or the supplementary material. The process of Semantic Clustering is a form of self-supervision.

**Self-Training**    Our process applies a simple self-training approach. Before training a model on the target domain, we create pseudo-labels for the target-domain images given the current model. To mitigate the influence of false pseudo-labels, we exclude labels from the self-training that yield a prediction-vector entropy exceeding the threshold $\beta \cdot \log K$, where $K$ is the number of classes and $\beta > 0$. Improving the quality and impact of the pseudo-labels for self-training requires implementing an iterative adaptation process. I. e., each adaptation step first needs to create new pseudo-labels before resetting the model's weights and (re-) starting the training.

### 3.2   EasyAdap: Assembling the Bricks

Figure 1 and Algorithm 1 in the supplementary material describe our training process. First, we perform the supervised training on the source-domain data with class-uniform sampling and a strong data augmentation to gain an initial model $M(0)$. Then, the process enters the domain adaptation loop. Each adaption step first (re-) creates the pseudo-labels using the current available model $M(n)$ (i. e., $M(0)$ in the first iteration). After resetting the model's weights (to weights pre-trained on ImageNet[31]), the adaptation step starts training a new model. We rebuild the list of samples regarding random and class-uniform samples in each epoch of the training. While iterating over the smaller (source- or target-domain) dataset, we sample as many batches as the larger dataset allows, i. e., we restart sampling from the smaller dataset until we finish processing the larger dataset. We compute the semantic segmentation loss for each training batch on the source- and target-domain data using the ground-truth and pseudo-labels, respectively. We use the features of the source-domain data to update the running average of the class centroids. The similarities between these centroids and the target-domain features yield the clustering loss. We update the model's weights using both loss functions.

## 4   Evaluation and Discussion

This section evaluates our approach, EasyAdap, regarding domain adaptation and domain generalization quality. To that, we evaluate its performance on the domain shift from synthetic to real-world data via adaptations from GTA5 [19] and Synthia [20] to Cityscapes [1], respectively.

We compare our approach against numerous top-performing approaches from the literature in Section 4.1. Since real applications require domain generalization instead of specific domain shifts, we

---

[5]The COCO format stores segmentation masks as polygons.

[6]These simple augmentations are available in the Tensorflow and PyTorch API.

[7]In contrast to [12], the method in [11] solves additional problems regarding memory and computational complexity due to the image-segmentation task.

[8]Note, that we define *clustering* as pulling elements towards defined cluster centers instead of finding a partition of the elements.

discuss and compare our approach regarding domain generalization against existing domain adaptation approaches in Section 4.2. Section 4.3 evaluates the impact of our data augmentation and the class-uniform sampling. In Section 4.4, we show the effects of the self-training and self-supervision. We conclude our evaluation with a discussion in Section 4.5

All of our experiments are based on an implementation of DeepLabV3+[9] with a WideResNet38 [32] backbone. We scaled all target-domain training images to match the size of the source-domain images. While we train our model on crops of $400 \times 400$ pixels in batches of 24 source- and 24 target-domain images, we validate it on the original sized images. The training process implements a stochastic gradient descent optimizer with momentum 0.9 and a weight decay of $10^{-4}$. While the source-only setups trained for 45 epochs, our iterative adaptation steps stopped early after 20 epochs. We decay the initial learning rate of 0.007 with a factor $1 - \frac{epoch}{epoch_{max}}$ after each epoch. We configure our domain adaptation approach as proposed in [11] except for the self-training threshold, which we now set to $\beta = \frac{1}{16}$ (see Section 3.1).

## 4.1 Comparison to Existing Approaches

Table 1 shows the performance of state-of-the-art unsupervised domain adaptation methods from GTA5 to the Cityscapes dataset. This table includes methods aligning distributions on the input space [6, 16], feature space [33, 34, 35], output space (e. g., via self-training [15, 27]), and hybrid methods [16, 17, 18] (see Section 2 for a description of this categorization).

Table 1: GTA5 to Cityscapes

| | road | sidewalk | building | wall | fence | pole | traffic light | traffic sign | vegetation | terrain | sky | person | rider | car | truck | bus | train | motorbike | bicycle | mIoU |
|---|---|---|---|---|---|---|---|---|---|---|---|---|---|---|---|---|---|---|---|---|
| source DLv2 | 75.8 | 16.8 | 77.2 | 12.5 | 21.0 | 25.5 | 30.1 | 20.1 | 81.3 | 24.6 | 70.3 | 53.8 | 26.4 | 49.9 | 17.2 | 25.9 | 6.5 | 25.3 | 36.0 | 36.6 |
| AdapSeg [13] | 86.5 | 25.9 | 79.8 | 22.1 | 20.0 | 23.6 | 33.1 | 21.8 | 81.8 | 25.9 | 75.9 | 57.3 | 26.2 | 76.3 | 29.8 | 32.1 | 7.2 | 29.5 | 32.5 | 41.4 |
| CyCADA [6] | 86.7 | 35.6 | 80.1 | 19.8 | 17.5 | 38.0 | 39.9 | 41.5 | 82.7 | 27.9 | 73.6 | 64.9 | 19.0 | 65.0 | 12.0 | 28.6 | 4.5 | 31.1 | 42.0 | 42.7 |
| CLAN [33] | 87.0 | 27.1 | 79.6 | 27.3 | 23.3 | 28.3 | 35.5 | 24.2 | 83.6 | 27.4 | 74.2 | 58.6 | 28.0 | 76.2 | 33.1 | 36.7 | 6.7 | 31.9 | 31.4 | 43.2 |
| APODA [36] | 85.6 | 32.8 | 79.0 | 29.5 | 25.5 | 26.8 | 34.6 | 19.9 | 83.7 | 40.6 | 77.9 | 59.2 | 28.3 | 84.6 | 34.6 | 49.2 | 8.0 | 32.6 | 39.6 | 45.9 |
| PatchAlign [37] | 92.3 | 51.9 | 82.1 | 29.252 | 25.1 | 24.5 | 33.8 | 33.0 | 82.4 | 32.8 | 82.2 | 58.6 | 27.2 | 84.3 | 33.4 | 46.3 | 2.2 | 29.5 | 32.3 | 46.5 |
| ADVENT [14] | 89.4 | 33.1 | 81.0 | 26.6 | 26.8 | 27.2 | 33.5 | 24.7 | 83.9 | 36.7 | 78.8 | 58.7 | 30.5 | 84.8 | 38.5 | 44.5 | 1.7 | 31.6 | 32.4 | 45.5 |
| sem. self-train. [11] | 82.5 | 43.9 | 76.4 | 31.7 | 24.7 | 45.2 | 45.6 | 22.5 | 87.1 | 30.9 | 82.6 | 71.0 | 41.8 | 86.5 | 28.0 | 27.8 | 0.01 | 25.5 | 27.3 | 46.4 |
| BDL [7] | 91.0 | 44.7 | 84.2 | 34.6 | 27.6 | 30.2 | 36.0 | 36.0 | 85.0 | 43.6 | 83.0 | 58.6 | 31.6 | 83.3 | 35.3 | 49.7 | 3.3 | 28.8 | 35.6 | 48.5 |
| CBST [27] | 91.8 | 53.5 | 80.5 | 32.7 | 21.0 | 34.0 | 28.9 | 20.4 | 83.9 | 34.2 | 80.9 | 53.1 | 24.0 | 82.7 | 30.3 | 35.9 | 16.0 | 25.9 | 42.8 | 45.9 |
| MRKLD [38] | 91.0 | 55.4 | 80.0 | 33.7 | 21.4 | 37.3 | 32.9 | 24.5 | 85.0 | 34.1 | 80.8 | 57.7 | 24.6 | 84.1 | 27.8 | 30.1 | 26.9 | 26.0 | 42.3 | 47.1 |
| FADA [35] | 91.0 | 50.6 | 86.0 | 43.4 | 29.8 | 36.8 | 43.4 | 25.0 | 86.8 | 38.3 | 87.4 | 64.0 | 38.0 | 85.2 | 31.6 | 46.1 | 6.5 | 25.4 | 37.1 | 50.1 |
| CAG [18] | 90.4 | 51.6 | 83.8 | 34.2 | 27.8 | 38.4 | 25.3 | 48.4 | 85.4 | 38.2 | 78.1 | 58.6 | 34.6 | 84.7 | 21.9 | 42.7 | 41.1 | 29.3 | 37.2 | 50.2 |
| Seg-Uncertainty [15] | 90.4 | 31.2 | 85.1 | 36.9 | 25.6 | 37.5 | 48.8 | 48.5 | 85.3 | 34.8 | 81.1 | 64.4 | 36.8 | 86.3 | 34.9 | 52.2 | 1.7 | 29.0 | 44.6 | 50.3 |
| CLST [10] | 92.8 | 53.5 | 86.1 | 39.1 | 28.1 | 28.9 | 43.6 | 39.4 | 84.6 | 35.7 | 88.1 | 63.9 | 38.3 | 86.0 | 41.6 | 50.6 | 0.1 | 30.4 | 51.7 | 51.6 |
| SAC [29] | 90.4 | 53.9 | 86.6 | 42.4 | 27.3 | 45.1 | 48.5 | 42.7 | 87.4 | 40.1 | 86.1 | 67.5 | 29.7 | 88.5 | 49.1 | 54.6 | 9.8 | 26.6 | 45.3 | 53.8 |
| Coarse2Fine [16] | 92.5 | 58.3 | 86.5 | 27.4 | 28.8 | 38.1 | 46.7 | 42.5 | 85.4 | 38.4 | 91.8 | 66.4 | 37.0 | 87.8 | 40.7 | 52.4 | 44.6 | 41.7 | 59.0 | 56.1 |
| ProDA [17] | 87.8 | 56.0 | 79.7 | 46.3 | 44.8 | 45.6 | 53.5 | 53.5 | 88.6 | 45.2 | 82.1 | 70.7 | 39.2 | 88.8 | 45.5 | 59.4 | 1.0 | 48.9 | 56.4 | 57.5 |
| source-only | 46.44 | 10.75 | 62.2 | 1.21 | 16.75 | 22.16 | 18.93 | 4.65 | 72.76 | 3.73 | 63.95 | 50.91 | 7.44 | 68.55 | 26.1 | 4.18 | 0 | 3.75 | 1.66 | 25.58 |
| source-only aug | 58.39 | 25.44 | 68.01 | 25.55 | 26.77 | 40.25 | 44.62 | 19.32 | 84.37 | 30.78 | 56.56 | 69.17 | 36.64 | 74.29 | 24.13 | 10.86 | 0.9 | 29.34 | 21.07 | 38.8 |
| source-only uni 50% | 57.87 | 32.03 | 58.53 | 23.62 | 25.06 | 42.38 | 45.17 | 28.22 | 83.36 | 26.02 | 81.01 | 70.16 | 40.24 | 80.07 | 20.05 | 15.95 | 1.02 | 32.47 | 23.32 | 41.4 |
| source-only uni 100% | 76.03 | 34.02 | 75.52 | 29.25 | 29.72 | 46.55 | 45.91 | 27.96 | 82.52 | 21.47 | 78.8 | 69.51 | 34.58 | 86.14 | 25.98 | 24.64 | 0 | 32.56 | 23.95 | 44.5 |
| Self-Train | 63.4 | 40.99 | 60.85 | 41 | 37.21 | 45.28 | 51.06 | 38.45 | 87.34 | 33.52 | 79.02 | 70.38 | 35.67 | 90.63 | 42.02 | 47.93 | 13.77 | 37.92 | 18.03 | 49.18 |
| Easy Adap | 87.84 | 56.1 | 80.68 | 37.21 | 40.12 | 49.39 | 55.04 | 47.18 | 86.87 | 39.54 | 85.35 | 69.93 | 42.13 | 90.65 | 52.12 | 61.45 | 0 | 42.13 | 46.39 | 56.32 |
| target | 98.01 | 84.41 | 92.07 | 49.66 | 59.59 | 64.43 | 68.76 | 78.22 | 92.36 | 63.49 | 94.3 | 82.17 | 62.3 | 94.82 | 80.36 | 85.76 | 79.74 | 65.99 | 76.93 | 77.55 |

Our method EasyAdap achieves near state-of-the-art performance in terms of mIoU on the GTA5 to Cityscapes domain change: We achieve 56.32% mIoU, which is comparable to the currently best approach ProDA with 57.5% mIoU. Similar to ProDA, we gain our improvement over other approaches from classes like *traffic light, traffic sign, fence, rider,* and *motorbike*, which are difficult to learn. Our method outperforms approaches that have similarities to our method (see Section 2) on this domain change. In particular, this includes CLST and CAG with 50.2% and 51.6% mIoU, respectively. Except for ProDA, the same holds for all hybrid methods. ProDA, however, is hard-to-tune due to its complex architecture (four structurally different training stages) and hyper-parameter-tuning. Advancing our method from [11] into an iterative domain adaptation process improves the results from 46.4% mIoU to 56.32% due to synergistic effects between our self-supervision and self-training. We further discuss these effects in Section 4.5.

Table 2 shows the performance of unsupervised domain adaptation methods from Synthia to the Cityscapes dataset. Without changing any hyper-parameters tuned for the GTA5 to Cityscapes domain change, our approach still performs well, ranking behind ProDA, SAC, and CLST, all of which are also well-performing in the adaptation from GTA5 to the Cityscapes dataset.

---

[9]Implementation based on https://github.com/NVIDIA/semantic-segmentation/tree/sdcnet

Table 2: Synthia to Cityscapes

| | road | sidewalk | building | wall* | fence* | pole* | traffic light | traffic sign | vegetation | sky | person | rider | car | bus | motorbike | bicycle | mIoU | mIoU* |
|---|---|---|---|---|---|---|---|---|---|---|---|---|---|---|---|---|---|---|
| source DLv2 | 64.3 | 21.3 | 73.1 | 2.4 | 1.1 | 31.4 | 7.0 | 27.7 | 63.1 | 67.6 | 42.2 | 19.9 | 73.1 | 15.3 | 10.5 | 38.9 | 34.9 | 40.3 |
| AdapSeg [13] | 79.2 | 37.2 | 78.8 | - | - | - | 9.9 | 10.5 | 78.2 | 80.5 | 53.5 | 19.6 | 67.0 | 29.5 | 21.6 | 31.3 | - | 45.9 |
| PatchAlign [37] | 82.4 | 38.0 | 78.6 | 8.7 | 0.6 | 26.0 | 3.9 | 11.1 | 75.5 | 84.6 | 53.5 | 21.6 | 71.4 | 32.6 | 19.3 | 31.7 | 40.0 | 46.5 |
| CLAN [33] | 81.3 | 37.0 | 80.1 | - | - | - | 16.1 | 13.7 | 78.2 | 81.5 | 53.4 | 21.2 | 73.0 | 32.9 | 22.6 | 30.7 | - | 47.8 |
| APODA [36] | 86.4 | 41.3 | 79.3 | - | - | - | 22.6 | 17.3 | 80.3 | 81.6 | 56.9 | 21.0 | 84.1 | 49.1 | 24.6 | 45.7 | - | 53.1 |
| ADVENT [14] | 85.6 | 42.2 | 79.7 | 8.7 | 0.4 | 25.9 | 5.4 | 8.1 | 80.4 | 84.1 | 57.9 | 23.8 | 73.3 | 36.4 | 14.2 | 33.0 | 41.2 | 48.0 |
| BDL [7] | 86.0 | 46.7 | 80.3 | - | - | - | 14.1 | 11.6 | 79.2 | 81.3 | 54.1 | 27.9 | 73.7 | 42.2 | 25.7 | 45.3 | - | 51.4 |
| FADA [35] | 84.5 | 40.1 | 83.1 | 4.8 | 0.0 | 34.3 | 20.1 | 27.2 | 84.8 | 84.0 | 53.5 | 22.6 | 85.4 | 43.7 | 26.8 | 27.8 | 45.2 | 52.5 |
| CBST [27] | 68.0 | 29.9 | 76.3 | 10.8 | 1.4 | 33.9 | 22.8 | 29.5 | 77.6 | 78.3 | 60.6 | 28.3 | 81.6 | 23.5 | 18.8 | 39.8 | 42.6 | 48.9 |
| MRKLD [38] | 67.7 | 32.2 | 73.9 | 10.7 | 1.6 | 37.4 | 22.2 | 31.2 | 80.8 | 80.5 | 60.8 | 29.1 | 82.8 | 25.0 | 19.4 | 45.3 | 43.8 | 50.1 |
| CAG UDA [18] | 84.7 | 40.8 | 81.7 | 7.8 | 0.0 | 35.1 | 13.3 | 22.7 | 84.5 | 77.6 | 64.2 | 27.8 | 80.9 | 19.7 | 22.7 | 48.3 | 44.5 | 51.5 |
| Seg-Uncertainty [15] | 87.6 | 41.9 | 83.1 | 14.7 | 1.7 | 36.2 | 31.3 | 19.9 | 81.6 | 80.6 | 63.0 | 21.8 | 86.2 | 40.7 | 23.6 | 53.1 | 47.9 | 54.9 |
| Coarse2Fine [16] | 75.7 | 30.0 | 81.9 | 11.5 | 2.5 | 35.3 | 18.0 | 32.7 | 86.2 | 90.1 | 65.1 | 33.2 | 83.3 | 36.5 | 35.3 | 54.3 | 48.2 | 55.5 |
| CLST [10] | 88.0 | 49.2 | 82.2 | 16.3 | 0.4 | 29.2 | 31.8 | 23.9 | 84.1 | 88.0 | 59.1 | 27.2 | 85.5 | 46.6 | 28.9 | 56.5 | 49.8 | 57.8 |
| SAC [29] | 89.3 | 47.2 | 85.5 | 26.5 | 1.3 | 43.0 | 45.5 | 32.0 | 87.1 | 89.3 | 63.6 | 25.4 | 86.9 | 35.6 | 30.4 | 53.0 | 52.6 | 59.3 |
| ProDA [17] | 87.8 | 45.7 | 84.6 | 37.1 | 0.6 | 44.0 | 54.6 | 37.0 | 88.1 | 84.4 | 74.2 | 24.3 | 88.2 | 51.1 | 40.5 | 45.6 | 55.5 | 62.0 |
| source-only | 8.6 | 11.58 | 32.01 | 1.45 | 0 | 30.25 | 18.55 | 9.1 | 74.57 | 68.74 | 56.63 | 8.16 | 66.98 | 12.05 | 3.41 | 9.69 | 26.31 | 29.24 |
| source-only aug | 55.14 | 30.54 | 69.12 | 4.69 | 0 | 40.35 | 25.78 | 23.55 | 80.15 | 76.93 | 61.92 | 21.59 | 40.78 | 18.6 | 17.05 | 27.6 | 37.13 | 42.21 |
| source-only uni 50% | 64.39 | 28.34 | 72.2 | 3.35 | 1.19 | 40.69 | 27.51 | 20.18 | 79.93 | 59.2 | 64.77 | 24.07 | 79.04 | 24.1 | 17.76 | 20.48 | 39.2 | 43.22 |
| source-only uni 100% | 79.0 | 33.18 | 68.75 | 3.16 | 0.38 | 42.07 | 25.79 | 26.05 | 78.67 | 76.76 | 61.28 | 24.76 | 80.72 | 26.64 | 18.73 | 28.94 | 42.18 | 48.41 |
| Easy Adap | 84.48 | 46.12 | 74.69 | 0.16 | 0.04 | 47.14 | 49.77 | 31.94 | 77.84 | 85.11 | 73.33 | 36.14 | 86.96 | 46.06 | 28.89 | 23.01 | 49.48 | 57.25 |
| target | 98.01 | 84.41 | 92.07 | 49.66 | 59.69 | 64.43 | 68.76 | 78.22 | 92.36 | 94.3 | 82.17 | 62.3 | 94.82 | 85.76 | 65.99 | 76.93 | 78.12 | 92.77 |

Table 3: Domain generalization: Methods trained on GTA5 (source-only) and methods adapted from GTA5 to Cityscapes tested on different real-world domains

| Model | Cityscapes | BDD | rain | fog | snow | night |
|---|---|---|---|---|---|---|
| source-only DLv2 [13] | 36.6 | 36.5 | 33.6 | 40.2 | 33.4 | 8.6 |
| source-only DLv2 [29] | 40.8 | 35.1 | 32.9 | 31.3 | 28.7 | 7.1 |
| source-only DLv3+ | 25.58 | 28.45 | 28.37 | 24.69 | 23.39 | 3.7 |
| source-only aug DLv3+ | 38.9 | 31.3 | 32.02 | 29.04 | 27.6 | 5.75 |
| source-only uni 100% DLv3+ | 44.06 | 36.77 | 33.27 | 35.75 | 28.28 | 7.63 |
| AdapSeg [13] | 42.4 | 37.4 | 30.8 | 35.4 | 27.9 | 7.4 |
| Seg-Uncertainty [15] | 50.3 | 35.7 | 35.9 | 41.4 | 37.4 | 14.0 |
| SAC [29] | 53.8 | 41.55 | 39.6 | 44.7 | 34.9 | 15.6 |
| ProDA [17] | 57.5 | 47.5 | 43.1 | 49.2 | 40.7 | 15.4 |
| EasyAdap | 56.6 | 46.69 | 43.02 | 48.56 | 38.87 | 14.38 |
| target (Cityscapes model) | 77.55 | 46.26 | 45.58 | 61.22 | 47.65 | 17.87 |

## 4.2 Domain Generalization

This section evaluates the domain generalization capabilities of our domain adaptation method and compares it with several other approaches. To that, Table 3 shows the performance of several models on the datasets Cityscapes, BDD [39], and four different domains of the dataset ACDC [40]. No model has seen samples of BDD or ACDC during the training. Furthermore, the source-only models did not see the Cityscapes samples during training.

BDD differs from the target domain of the adaptation (Cityscapes) by containing a diverse set of weather and lighting conditions and by being recorded across the USA instead of Germany. ACDC, which was recorded in Europe, defines the specific partitions *rain, fog, snow,* and *night*. Hence, regarding the models trained on GTA5 (and Cityscapes without ground-truth), these datasets include domain shifts regarding the environment, weather, and sensors.

The improvements of our source-only models due to data augmentation and additional class-uniform sampling also transfer to the unseen domains of BDD and ACDC. Considering the DeepLabV2 [29, 13] and DeepLabV3+ source-only models, there is no clear superior model. Note that the target model (trained on the Cityscapes dataset) yields similar results on BDD as the adaptation approaches ProDA and EasyAdap from GTA5 to Cityscapes. Apart from that, the target model always outperforms the domain adaptation approaches significantly. While ProDA leads the board of domain adaptation approaches, EasyAdap achieves very close results. Furthermore, EasyAdap outperforms the other domain adaptation approaches (see table 3).

## 4.3 Source-Only Training

The quality of our source-only training is crucial for the later creation of the first pseudo-labels. To study the design decision of our source-only training, Table 1 and 2 show the performance of different source-only models: trained without any augmentation (source-only); trained with color jitter, Gaussian blurring, and random scaling (source-only aug); and trained with augmentation and class-uniform sampling (source-only uni 50% and source-only uni 100%). On both datasets, GTA5

and Synthia, the augmentation improves the plain source-only model's quality by over $10\%$ mIoU. An additional class-uniform sampling improves the model's quality further by over $5\%$ mIoU. Sampling the complete batch via class-uniform sampling (source-only uni $100\%$) outperforms a mixed batch of random and class-uniformly sampled images (source-only uni $50\%$).

## 4.4 Impact of Adaptation Features

We conducted several experiments to study the impact of EasyAdap's features on the model's quality. Figure 2 shows the best mIoU values on the validation set per adaptation step, which consists of $N_{train} = 20$ training epochs (see Algorithm 1). We successively enabled self-training, class-uniform sampling[10], and the semantic self-supervision on the target domain (see Section 6.1 and 3.1). Figure 2 shows that a self-training yields similar improvements as a combination with an additional class-uniform sampling on the target domain. A further self-supervision shows a steeper learning curve in the first few adaptation steps and yields a more durable learning behavior. I. e., the two learning processes without self-supervision stagnate after four adaptation steps, but the model trained with additional self-supervision still increases in quality.

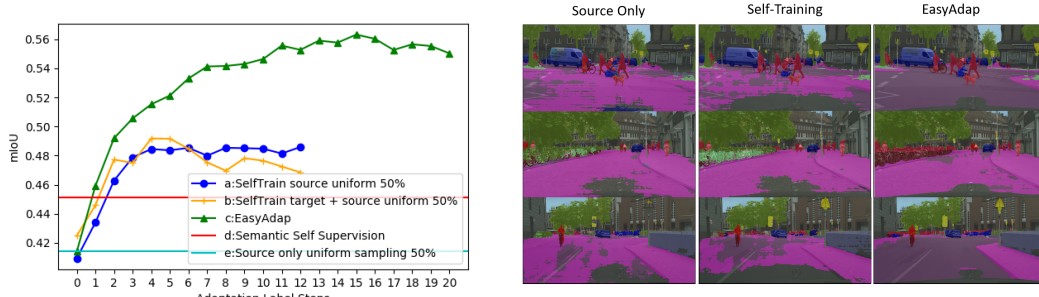

Figure 2: Left: Best mIoU per adaptation step; self-training (a); self-training+uniform sampling on target domain (b); self-training+uniform sampling+semantic self-supervision (c); uniform sampling+semantic self-supervision (d); uniform sampling source only (e). Right: Qualitative examples

## 4.5 Discussion and Limitation

*Does domain adaptation transfer knowledge?* Table 3 shows that the target model trained on Cityscapes (*Cityscapes model*) achieves similar performance on BDD as the ProDA and EasyAdap models adapted to the Cityscapes dataset. While the Cityscapes model has to overcome the domain change from sunny to rainy weather included in BDD, the adaptation methods need to overcome the domain change from synthetic to real-world. The Cityscapes model also achieves similar performance on ACDC's rain domain as the ProDA and EasyAdap models. In this case, the target domain consists of rainy images only, which challenges the Cityscapes model even more, while the adapted models from GTA5 seem to play out their advantage of having seen synthetic rainy images during their training. On the other hand, the Cityscapes model outperforms the adapted models on the foggy and snowy domains, which are unseen for all adapted models. We think that the additional domain gap from the synthetic to the real world for the adapted models is the reason for the performance drop of the adapted models compared to the Cityscapes model. Hence, the adapted models show the behavior of knowledge transfer (here, rainy images) from synthetic to real-world domains. This observation supports approaches adapting from synthetic to real-world domains. Moreover, it supports the demand for richer synthetic datasets regarding different domains, such as sensor, weather, lighting, and environmental domains.

*How to support domain generalization through adaptation?* Since a domain adaptation to every possible encountered domain is infeasible in real-life applications, this work explicitly addresses and

---

[10]From here on, we always apply class-uniform sampling on the source domain.

studies domain generalization effects of unsupervised domain adaptation methods, including our own method EasyAdap. Section 4.2 and Table 3 show stronger generalization effects for some of the adaptation methods. In particular, SAC, ProDA, and EasyAdap show significant improvements in the unseen domains of BDD and ACDC. These methods apply self-training while AdapSeg does not, and Seg-Uncertainty only applies self-training without improving the pseudo-labels through recreation. These observations open the question of whether self-training, which mimics a supervised training on both domains, yields these strong generalization effects.

*What are the reasons for synergistic effects?* The experiments in Section 4.4 and Figure 2 show a synergistic effect since enabling self-training and self-supervision gains $15\%$ mIoU while enabling only one of the mechanisms gains only $8\%$ and $4\%$, respectively. Hence, simultaneously enabling both mechanisms gains another $3\%$ atop each single mechanism. We argue that there is a synergy between self-training and semantic self-supervision, i.e., they support each other. On the one hand, self-training aims to mimic a supervised training on the source and target domain, which helps the training process to generalize upon both domains. This generalization effect yields a better alignment of the feature distribution in both domains, which in turn helps the self-supervision to attract the target-domain features to the correct source-domain class centroids. On the other hand, the self-supervision reduces the number of correct but ignored pseudo-labels by sharpening the feature space and therefore reducing the uncertainty of correctly classified pixels.

*How does the sampling strategy affect generalization?* Sections 4.2 and 4.3 show that our source-only models generalize well to unseen real-world data. We argue that combining a strong data augmentation with a class-uniform sampling improves the domain generalization by avoiding over-fitting. On the one hand, seldom classes gain weight in the training by oversampling, but the strong augmentation avoids overfitting. On the other hand, large-area classes such as vegetation and terrain lose weight compared to seldom classes, which again avoids overfitting the training data.

## 5   Conclusion

We propose the easy-to-use unsupervised domain adaptation method EasyAdap with near state-of-the-art performance based on a self-supervision and self-training strategy. Our evaluation shows that the quality of our method comes from a combination of improvements, including preliminary training on the source domain with increased domain generalization capabilities due to strong data augmentation and a class-uniform sampling; a self-training that yields strong domain generalization capabilities; and a synergy of our self-supervision and self-training. Our evaluation includes an extensive discussion of domain adaptation methods at hand regarding knowledge transfer and domain generalization capabilities. Our discussion includes several research questions regarding domain adaptation and domain generalization of training and adaptation methods.

## Acknowledgement

The research leading to these results is funded by the German Federal Ministry for Economic Affairs and Climate Action" within the project "KI Delta Learning" (Förderkennzeichen 19A19013K). The authors would like to thank the consortium for the successful cooperation.

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
