# OpenReview forum: "Domain Adaptation and Generalization: A Low-Complexity Approach"
_robot-learning.org/CoRL/2022/Conference — CoRL 2022 Poster_

### Official Review · Reviewer_vMEa · 2022-07-25

**Originality:** Poor
**Technical Quality:** Very Good
**Clarity Of Presentation:** Good
**Impact:** 3

**Recommendation:**

Weak Accept: I recommend accepting the paper, but will not argue for my recommendation if the majority of other reviewers have a different opinion.

**Summary:**

The authors present a method for unsupervised domain adaptation. Many methods already exist which perform domain adaptation by aligning the distributions of the source and target data in the input space, feature space, output space, or a combination of them. The authors focus on the feature space and provide a straightforward method which uses multiple iterations self-training and semantic clustering to perform well on a target domain. The authors compare with many other methods and show strong performance.

**Issues:**

Content Issues:
- L133: How is this work inspired by [11]?
- Sec 3. I am confused as to why the methods are so lightly discussed in the body of the paper as the authors mention EasyAdap as a key contribution of the work.
- The authors claim their methods are easy to implement, however;
	- The authors do not formalize the domain adaptation problem well in Sec. 3. They offload much of the formalization to another paper [10] and only elaborate slightly in the appendix. E.g., there is a lack of information about the sampling and augmentation done in algorithm 1.
	- The importance of good source training seems to limit the applicability of the proposed method. It may take someone a long time to determine a good set of augmentations for a single problem. I see this counter to the idea of easy implementation.
	- The authors do not provide code for reproducing their results.
- Please discuss the limitation of strong source training on the performance of the proposed domain adaptation method.
- Section 4.3. Some of the methods in Table 1 and 2 are not mentioned anywhere else in the paper, e.g. PatchAlign [36], APODA [35]. Additionally, methods such as CLAN [32] and FADA [34] are left vaguely as domain adaptation methods which align the feature space. The authors should expand on the explanation of many of comparison methods. If necessary, this could be in the appendix.
- Algorithm 1, L6: I am confused as what is meant by “reset(M)”. Do you reset the weights of M to untrained, and then need to do supervised source-only training again? If so, why is the source-only training necessary again? If not, what is M reset to and I am still not understanding why is it reset?


Editorial issues:
- The in-line reference, equation, and section hyper-links are not working for me.
- I am not certain the authors are using “cf.” correctly. E.g. in L113-L114: “We initialize the process with a training on the source domain (cf. Section 3.1).” I would interpret “(cf. Section 3.1)” as “compare to Section 3.1”, but it seems like they mean “see Section 3.1”. I believe the authors make this mistake frequently throughout the paper.
- Algorithm 1, L18 - L22: Equations references are not referenced correctly
- In Table 1, method “ADVENT” has reference [13] and in table 2 it has reference [35]. Is the [35] reference a mistake? If so please ensure the methods in tables are cited correctly.

**Quality Of The Limitations Section:**

Additional details required

**Reviewer Expertise:**

3: The reviewer is fairly confident that the evaluation is correct

**Robotics Focus:**

Highly relevant to robotics but no hardware experiments

**Strengths And Weaknesses:**

Strengths:
- The authors compare to a wide variety of other methods and show strong performance in multiple settings.
- The authors provide an interesting and comprehensive discussion, talking about the reasons domain adaptation works and the impacts of factors such as the sampling strategy and differing target/source domains.

Weaknesses:
- The main contribution of the paper EasyAdap is a very slight extension of the method in [10]. EasyAdap runs multiple iterations of self-training and semantic clustering instead of a single iteration in [10], and I do not see this as particularly novel.
- I am not convinced the methods are easy to use. Specifically, for good performance, potentially expensive and complicated data augmentation may be required. E.g. How does someone who wants to use the methods decide on how to augment the data? How do they know when there is enough augmentation? This would significantly complicate the implementation of the methods.
- Even if the method is easy to implement, the authors do little work in ensuring that others can use their methods. E.g. there is no code to reproduce the results in the paper, the body of the paper does not explain EasyAdap thoroughly, and the methods are missing other important details which would allow for the use and understanding of the methods (e.g. precise details about the source training steps, and why the weights of source-trained model are reset).

**Summary Of Recommendation:**

The authors compare their method to a large variety of methods and show strong performance in multiple settings. However, the presentation needs a lot of work and there is minimal algorithmic contribution since the method is so similar to [10]. I think the paper would need a significant amount of re-writing and I cannot currently recommend acceptance.
I do hope the authors address the comments and issues in the revision or submit a revised version of the paper in the future.

---

**Update after revisions and replies**: The authors have improved the paper and clarified a few important issues. The focus has shifted slightly away from a broadly easy to use method to a lower-complexity method, which I think improves the story of the paper. I also appreciate that the authors are willing to include a link to code if the paper is published. My main reservations are the similarity to [10] and the potential difficulty in deciding how to augment the training dataset. However, the performance of EasyAdap provides evidence that the extension from [10] improves performance significantly. Additionally, code and concrete implementation examples including a good set of augmentations provides a good starting point if others want to use this method. I will therefore raise my recommendation.

---

> ### Author Response · Authors · 2022-08-18
> **Response to Reviewer vMEa**
>
> We would like to thank the reviewer for the thorough examination of our paper and his helpful feedback. In the following we would like to address the issues.
>
> **Importance of a good source training for the applicability**
> In this paper we have analyzed our source only
> training strategy for the synthetic to real world domain shift.
> We have shown for the important datasets Syhnthia and
> GTA5 the uniform sampling and the color jittering introduces a strong generalization effect. Hence at least the
> applicability for the sythetic to real use case is given. If
> however we were to not use a source only training without
> any generalization strategy the generation of pseudo labels
> would indeed be harder.
> For a different field then the synthetic to real adaptation in autonomous driving other augmentations might be even better however our experience would lead us to assume that our augmentation would
> even perform well for e.g. sensor domain shifts.
>
> **Some methods in the table 1 and 2 are not or vaguely introduced in the related work**
> We limited the related work section given the space. We are happy to include it in the appendix as the Reviewer suggest.
>
> **lack of information about the sampling and augmentation** We are happy to provide more details of our augmentation pipeline and the sampling either in the appendix or the section 3.1.
>
> **What is meant by “reset(M)”?** With “reset(M)” we
> mean resetting the segmentation model M to the initialization weights, which in this case are taken from a imagenet
> pre-training. This is part of the iterative domain adaptation loop. To create the first model M(0) we train on the
> source domain. The model M(0) infers pseudo labels on
> the target domain. After that we can compute the cross entropy loss on the source and the target domain. We train
> the next model M(1) based on the source domain labels, the
> target domain pseudo labels and the clustering loss. After
> 20 epoch an adaptation loop is finished. We hence recreate
> the target domain pseudo labels again, which are now of a
> better quality. We now can reset the Model again and start
> the next adaptation iteration to compute M(2). This process
> is described in section 3.2.
>
> **Why is the method easy to use?** The synergistic effect between self-training and self-supervision allows us to
> achieve near state of the art performance with only two loss
> functions. Other methods like ProDA [16] and Coarse2Fine
> [15] combine many loss functions and structurally different stages. If the number of elements increases so does
> the need for finetuning the hyperparameters. The ProDA
> Method combines multiple loss functions and 4 structurally
> different training stages: Stage 1 is a warm up phase with
> an adversarial training in the output space; Stage 2 consists
> of a combination of self-training, the additional symmetric entropy loss function, and an adversarial structure learn-
> ing; Stage 3 combines self-supervision and self-distillation;
> and Stage 4 conducting knowledge distillation. The Method
> Coarse2Fine [15] is comprised of three structurally different straining stages, as well. In Stage 1 they utilize a style
> transfer to the target domain to create a source domain training with the target domain style; In Stage 2 they apply
> self-Training on the target domain, a triplet loss to increase
> the feature space difference between different classes and
> a consistency loss on the target domain; In Stage 3 they
> repeat the Step 2 whilst optimizing the pseudo labels for
> self-training. The two examples show that it is common for
> the top performing unsupervised domain adaptation strategies to be comprised of a variety of sub elements. The hyper parameters of such elements need to be fine tuned and
> balanced with each other. The reviewer is right in pointing out our method falls into the
> hybrid category. Compared to e.g. ProDA and Coarse2Fine
> however our process is a lot less complex.
>
> **Editorial issues** Good hint, the term “c.f.” will be replaced. ”Algorithm 1, L18 - L22: Equations references are
> not referenced correctly”: Thanks for pointing this out, we
> will fix it. ”In Table 1, method “ADVENT” has reference
> [13] and in table 2 it has reference [35]”: We will fix it.
>
> **Domain Generalization** We would like to point the attention to another aspect of the paper. We analyzed if domain adaptation introduces domain generalization (sections 4.4/4.5). Table 3 shows the results of networks that were adapted from the synthetic GTA5 domain to the real world Cityscapes domain. The networks are tested on real world domains that were not seen before. This introduces the research question if the domain adaptation from synthetic to real world data introduces a generalization effect to the whole real world domain. This is important since adapting to every real world sub domain would be infeasible. Introducing this research question hence is of particular relevance for robotic systems and to our knowledge is done in this paper for the first time.

---

> > ### Author Response · Authors · 2022-08-26
> > **Additional comments**
> >
> > We would like to post some additional comments w.r.t. the review.
> >
> > **Editorial issues:**
> > Please view the revised version of our paper. The chages are marked in blue.
> > We have replaced the use of c.f. with the more appropriate "see".
> > In Algorithm 1 we have fixed the issue of referencing the equations correctly.
> > In Table 1 the reference of ADVENT is correct, now.
> >
> > **Algorithm 1: reset(M)**
> > In line 156-157 we now describe more thoroughly what we mean by resetting "M".
> >
> > **Information about the sampling and augmentation**
> > Apart from the information given about the augmentation pipeline in section 3.1. (lines 123 - 133 in the revised paper), we have added a footnote at page 4 to point out that we used the normal torch implementation of the Gaussian blurring, color jittering, and random scaling.
> > In section 4.5 (lines 281-286 of the revised paper) we discuss how the augmentation strategy effects the generalization.

---

### Official Review · Reviewer_fxUx · 2022-07-25

**Originality:** Good
**Technical Quality:** Very Good
**Clarity Of Presentation:** Very Good
**Impact:** 2

**Recommendation:**

Weak Reject: I recommend rejecting the paper, but will not argue for my recommendation if the majority of other reviewers have a different opinion.

**Summary:**

The authors propose EasyAdap method. The consists of the following three modules:
(a)Source Training: use supervised learning method to train on data from source domain. It is designed to pretrain a good enough feature extractor and classifier for the following module.
(b)Semantic Clustering: this module is used to evaluate the similarity between target domain and source domain.
(c) Self training: pseudo label for data from target domain is generated and added for supervised training.

The authors evaluate their method on GTA5, Synthia, and Cityscapes dataset, and has a similar performance as the best SOTA method.
The authors include further discussion after their experiments.

**Issues:**

1. Figure 1 is extremely hard to read. It is very hard for someone who is not familiar with the specific dataset to understand what the pseudo-labeling is doing. In addition, it is also not clear what is happening in the self-supervision module.
2. The authors should add more mathematical equations in their paper to make it more clear about what their method is doing. I noticed that there is some equations in the supplementary materials, but it would be nice if it could also be included in the main text.
3. The authors should propose more qualitative analysis, e.g., what the pseudo labelling is doing. How the performance change with the iterative training.

**Quality Of The Limitations Section:**

Limitations are not well addressed

**Reviewer Expertise:**

3: The reviewer is fairly confident that the evaluation is correct

**Robotics Focus:**

Relevant but unlikely to deploy to hardware in near future

**Strengths And Weaknesses:**

Strengths:
1. The design of the framework is clearly written.
2.  The authors have included complete ablative analysis in their experiments

Weakness:
1. EasyAdap doesn't seem to have a better performance than the SOTA baseline, e.g., ProDA
2.  I think the experiments in the paper do not give an intuition why their methods could have a good performance over lots of baselines. For example, why do we need to add target domain data into the supervised training set and train again? Without such an intuition, I am not clear about how much distribution shift EasyAdap can handle. The authors only listed numbers and stats, but didn't explain that is happening inside their algorithm. For example, It would be very benefitial if the authors show some examples of the pseudo labels and how it improves during the iterative training procedure.


**Summary Of Recommendation:**

The authors propose EasyAdap with great clarity and have sufficient experiments. However, the authors have to explain more about why and how their method helps. And another major issue is that their method doesn't beat the SOTA baseline.

---

> ### Author Response · Authors · 2022-08-18
> **Response to Reviewer fxUx**
>
> We would like to thank the reviewer for the thorough examination of our paper and his helpful feedback. In the following we would like to address the issues.
>
> **Understandability** We understand the reviewers concern
> w.r.t. the figure 1. We will release an updated version of this
> figure. Due to the space constraints we have put the equations and the pseudo code in the supplementary material.
> Similarly qualitative results were difficult to fit in the main
> text. How the pseudo labels evolve however is briefly presented in figure 1. Qualitative results of EasyAdap are pre-
> sented in figure 2. In the supplementary material, we show
> the generalization effect after the adaptation from GTA5 to
> Cityscapes on camera data that was taken from our own autonomous vehicles. Hence further analyzing the applicability of the method.
>
> **Does not beat the SOTA baseline** We understand the the
> reviewers concern. However we would like to point out that
> we introduced a structurally new way of achieving nearly
> the same performance. This is important for two reasons.
> On the one hand one could think of combining the two approaches to achieve even better results and on the other hand
> follow up paper of this line of research could deliver further
> promising results. Other than this we would like to point
> out that our approach is less complex.
>
> **Reason for good performance over lots of baselines?**
> We would like to point out that no manual labeled data
> was used for the target domain. Our approach in purely
> unsupervised. The annotation of the target domain is done
> based on pseudo labels that are generated based on inferences of the adapted model itself. After an adaptation circle is finished the model is reinitialized with image net pretraining weights. This is done to prevent overfitting. We
> train again, since the pseudo labels are generated with the
> current model and hence are of a better quality. In section
> 4.5 (”What are the reasons for synergistic effects”) we describe the synergistic effect between the self-training and
> the self-supervision. This synergistic effect allows for a
> training strategy that is low in complexity but achieves a
> good performance
>
> **How much distribution shift EasyAdap can handle?**
> This indeed is an interesting question. Generally speaking
> the synthetic to real world domain shift is within robotics/
> autonomous driving one of the hardest and most relevant
> shifts. We show for the two most important dataset adaptations scenarios that EasyAdap is capable to perform well.
> As the generation of synthetic data get’s better EasyAdap
> should perform even better on synthetic datasets that will
> be released in the future. Other shifts that include e.g. different sensors or environment conditions cause less of a
> performance drop. Since the source only training will already produce good pseudo labels and the feature distributions of source and target domain are well aligned, as well, the
> self-training and the feature clustering are likely to perform
> welL
>
> **Domain Generalization**
> Apart from the issues we would like to point the attention to another aspect of the paper. In this paper we not
> only presented a new method but analyzed if unsupervised
> domain adaptation introduces domain generalization (sections 4.4 and 4.5). Table 3 shows the results of networks that
> were adapted from the synthetic GTA5 domain to the real
> world Cityscapes domain. The networks are tested on real
> world domains that were not seen before. This introduces
> the research question if the unsupervised domain adaptation
> from synthetic to real world data introduces a generalization
> effect to the whole real world domain. This is especially important since adapting to every real world sub domain would
> be infeasible. Introducing this research question hence is of
> particular relevance for robotic systems and to our knowledge is done in this paper for the first time. Another interesting aspect we covered in the discussion is the question,
> whether knowledge of the synthetic domain can be transferred to the real world domain.

---

> > ### Author Response · Authors · 2022-08-26
> > **Additional comments**
> >
> > We would like to post some additional comments w.r.t. the review
> >
> > **Figure 1 is extremely hard to read**
> > We have updated the figure to make it more understandable.
> > You can view it in the revised version of the paper we uploaded, today.

---

### Official Review · Reviewer_geXK · 2022-07-31

**Originality:** Good
**Technical Quality:** Good
**Clarity Of Presentation:** Good
**Impact:** 3

**Recommendation:**

Weak Accept: I recommend accepting the paper, but will not argue for my recommendation if the majority of other reviewers have a different opinion.

**Summary:**

The paper proposes an unsupervised domain adaptation method for semantic segmentation named "EasyAdap". The method is evaluated on autonomous driving datasets, with the goal of transferring a semantic segmentation model for traffic scene segmentation from a synthetic to a real (e.g. Cityscapes) domain. The method studies both domain adaptation (unlabeled target domain data available at training time) and the domain generalization setting (no target domain data available at training time). The method's key idea is to combine feature-space distribution alignment (similar classes have similar features across domains) and output-distribution alignment (model produce similar output distributions on source & target domain). Feature-distribution alignment is addressed through self-supervision using a clustering-like loss, output-distribution alignment through self-training on the target domain using pseudo-labels generated on the source domain.

**Issues:**

General:
  - Make iterative self-training and self-supervision + its synergistic effect the central theme of the paper
  - Drop or properly defend the "easy" claim of EasyAdap

Methods:
  - Improve the algorithm explanation by using more consistent and precise language (see explanation under strengths)
  - Explain the point about smaller vs. larger dataset priority

Experimental section:
  - Reorder sections from key result to ablation study (4.3, 4.4, 4.1+4.2)
  - Baseline difference. Explain why the paper improves upon it's predecessor [10].

Misc:
  - Related work. Repeat here that the proposed method falls into the hybrid category (it does, right?)
  - Footnote 1: Fix or substantiate this claim with references

**Quality Of The Limitations Section:**

Additional details required

**Reviewer Expertise:**

4: The reviewer is confident but not absolutely certain that the evaluation is correct

**Robotics Focus:**

Highly relevant to robotics but no hardware experiments

**Strengths And Weaknesses:**

## Strengths
- The paper is mostly well-written
- The method is technically sound and mostly understandable
- The results show that the proposed methods reaches near state-of-the-art performance, and provide a thorough ablation study. I particularly liked the discussion on the interplay between self-training and self-supervision.
- The related work section gives a very good overview of SoTA approaches to domain adaptation.


## Weaknesses

### Writing

The paper is mostly well-written, but I have some concerns regarding the clarity of the technical explanation.

#### Motivation
- I would like to see a better motivation for the way the method is put together in the introduction. The paper makes a convincing argument that self-training and self-supervision seem to have synergistic effects (discussion section). I would suggest to make this the main hypothesis of the paper, introduce and motivate it in the introduction, and use it as a guiding principle through the explanation of the paper.

#### Algorithm explanation
- While reading, I repeatedly got confused by self-training vs. self-supervision - these terms are very similar and easy to confound. Also, to me the clustering loss could equally be called an "unsupervised" loss. I would suggest to very clearly articulate (maybe even in a table) that the feature-distribution adaptation is achieved through self-training and the output-distribution adaptation through self-supervision.

- 3.2 I think this paragraph could be improved by referring more explicitly to Figure 1, and by being more precise and consistent about the wording. For example, I would make more explicit which model is trained on which domain (e.g. that in 1 only source domain) with which labels (in 1 source domain comes with "free" labels, right?). Also, I got confused about "smaller vs. larger" dataset in line 156. This seems to be an important detail, but what's the purpose of this, why is it important?

#### Experimental results
- Section ordering. I would suggest to reorder the sections. The paper currently starts with ablation studies (4.1, 4.2), then moves on to domain adaptation results (4.3) and then domain generalization (4.4). I would start with the most important results first (4.3) and also rename the section to "Domain Adaptation", then 4.4, and followed by 4.1 and 4.2 where I would suggest using "Ablation study" in the titles.

- Baseline difference discussion [10]. The paper explains that it extends the two-step method [10], which is also included in the table. I would have expected a section explaining the performance differences between the two methods as part of the experimental results, and analyze how and why the iterative setup improves results.

- Clarity of Table 1 and Table 2. I would like to see a better explanation of tables 1 and 2. I was struggling to understanding initially what the "source" vs. "target" rows where and the latter is not explained in the paper (I figured it's the upper bound training on target domain data - but how is it trained?).

- It would be nice to structure explicitly say which baselines are feature-based, output-based and hybrid

#### "Easy"

I have concerns regarding claiming the method is "easy". There is no concentrated place in the paper where the claim is justified. The abstract mentions that the method is "easy to implement", which  I find a very subjective claim - also does it matter in times where models are on github? The introduction claims "easy to reproduce", which is not explained at all -- again, if you fix random seeds and use the same code, isn't reproducibility a no-brainer? I would find "easy to use" (no extensive hyperparameter tuning, etc.) the most convincing interpretation of "easy", but no results nor discussion is provided in the paper (apart from stating that the same hyperparameters where used for domain adaptation and generalization).

Also, the paper seems to contradict itself, e.g. when claiming using a "sophisticated" model initialization in line 123.

However, I think the paper still makes a contribution even if it was not "easy". I would suggest to focus on the iterative combination of self-training and -supervision and makes this the central theme and title of the paper.

### Misc

- Related work. It would be great to repeat here that the proposed method falls into the hybrid category (it does, right?)
- Footnote 1: This claim is wrong as it makes a general statement about any pixel-wise domain. The authors should clarify on what dataset these 90 minutes apply and provide a source. I find 90 minutes extremely hard to believe. I would claim trained annotators using good tools can label complex Cityscapes scenes within minutes.


**Summary Of Recommendation:**

The paper tackles an important problem and shows convincing results and progress. The paper is mostly understandable but I would suggest to improve the clarity of the technical writing and the experimental section. Finally, I am challenging calling the method "easy" since I do not see convincing evidence for claiming this to be the case nor the key feature of the method.

---

> ### Author Response · Authors · 2022-08-18
> **Response to Reviewer geXK**
>
> We would like to thank the reviewer for the thorough
> examination of our paper and his helpful feedback. In the
> following we would like to address the issues.
>
> **Why do we call the method ”Easy”Adap?**
> As the reviewer suggests easy is indeed interpreted as easy-to-use in
> this context. As the reviewer points out correctly in this
> work we generated a strong synergistic effect between the
> self-training and the self-supervision. This synergistic effect allows us to achieve near state of the art performance
> with only two loss functions. The ProDA [16] method on
> the other hand combines multiple loss functions and 4 structurally different training stages: Stage 1 is a warm up phase
> with an adversarial training in the output space; Stage 2
> consists of a combination of self-training, the additional
> symmetric entropy loss function, and an adversarial structure learning; Stage 3 combines self-supervision and self-
> distillation; and Stage 4 conducting knowledge distillation.
> The Method Coarse2Fine [15] is comprised of three training stages. In Stage 1 they utilize a style transfer to the
> target domain to create a source domain training with the
> target domain style; In Stage 2 they apply self-training on
> the target domain, a triplet loss to increase the feature space
> difference between different classes and a consistency loss
> on the target domain; In Stage 3 they repeat the Step 2 whilst
> optimizing the pseudo labels for self-training. The two examples show that it is common for the top performing unsupervised domain adaptation strategies to be comprised of
> a variety of sub elements. The hyper parameters of such elements need to be fine tuned and balanced with each other. The reviewer is right in pointing
> out our method falls into the hybrid category. Compared
> to methods like e.g. ProDA and Coarse2Fine however our process is a
> lot less complex. As the reviewer suggests easy is indeed
> interpreted as easy to use in this context
>
> **90 minutes annotation time**
> In the original Cityscapes paper the following was stated: ”Annotation and quality control required more than 1.5 h on average for
> a single image.":http://download.visinf.tu-darmstadt.de/papers/2016-cvpr-cordts-cityscapes-preprint.pdf
>
> **Experimental section:**
> (1) How does the method improve
> upon the predecessor [10]? Apart from the synergistic effect
> that is discussed in section 4.5 the improvements come from
> the iterative process. Since the semantic clustering leads
> to a model that delivers better performance on the target
> domain, the pseudo labels that are generated are improved,
> also. Given improved pseudo labels we then again can train
> a better model on the target domain. (2) We are open to reorder
> the experiment subsections, however we would suggest a
> slightly different order: (4.3, 4.1+4.2, 4.4). That is because
> in section 4.4 we introduce the research question whether
> domain adaptation introduces domain generalization
>
> **Domain Generalization is important for Robotic learning:**
> Apart from the issues we would like to point the attention to another aspect of the paper. In this paper we not
> only presented a new method but analyzed if unsupervised
> domain adaptation introduces domain generalization (sections 4.4 and 4.5). Table 3 shows the results of networks that
> were adapted from the synthetic GTA5 domain to the real
> world Cityscapes domain. The networks are tested on real
> world domains that were not seen before. This introduces
> the research question if the unsupervised domain adaptation
> from synthetic to real world data introduces a generalization
> effect to the whole real world domain. This is especially important since adapting to every real world sub domain would
> be infeasible. Introducing this research question hence is of
> particular relevance for robotic systems and to our knowledge is done in this paper for the first time. Another inter-
> esting aspect we covered in the discussion is the question,
> whether knowledge of the synthetic domain can be transferred to the real world domain.

---

### Meta-Review · Area_Chair_5aQj · 2022-08-12

**Recommendation:** Accept (Poster)
**Confidence:** 4

**Metareview:**

The authors proposed an unsupervised domain adaptation method for semantic segmentation based on self-supervision and self-training strategies. The method was evaluated on autonomous driving datasets, and showed strong performance compared to other methods.

While most of the reviewers agree that the method show convincing results, with strong performance in multiple settings compared to other methods and with a comprehensive ablation study, they brought up three important areas to improve:

(1) Better clarity is needed for the main algorithm, such as better explanations of self-training vs self-supervision and Fig 1.

(2) The paper lacks explanation of why it can outperform the baselines especially its predecessor [10].

(3) The claim of the method is "easy" is hard to justify.

During the reviewer discussion, most reviewers agreed the authors have addressed the weakness in the revised paper, and the story lands better after shifting from easy-to-use method to a lower-complexity method. Reviewers agree that this paper is of interest to the community, but has limited technical contribution. For the above reason, we recommend accepting the paper as poster.

---

> ### Author Response · Authors · 2022-08-26
> **EasyAdap: Domain Adaptation and Generalization**
>
> We would like to thank the Area Chair for pointing out the main areas of the paper that should be improved.
> We have addressed the three points in the revised version of the paper that we have uploaded, today,
>
> **(1) Better clarity**
> We have updated the figure 1 to be more comprehensive. Hence the main algorithm in section 3.2 is easier to understand.
> Additionally we have updated the pseudo code is the supplementary material.
> Apart from that we have clarified the confusion of the terms self-training vs self-supervision (line 142-143 in revised paper).
>
> **(2) Explanation of why it can outperform the baselines**
> In line 198-200 we have introduced an explanation why our model outperforms [10].
> Apart from that, the synergistic effect between self-supervision and self-training that allows us to outperform the baselines is discussed in section 4.5 (line 270-280 in the revised version of the paper)
>
> **(3) The claim of the method is "easy" is hard to justify**
> We have addressed this issue in two ways in the revised version of the paper.
> On the one hand we have dropped the term from the title as reviewer "geXK" suggested.
> We rather interpreted the statement as low-complexity and easy to use.
> We hence changed the title to "Domain Adaptation and Generalization: A Low-Complexity Approach" in the revised version of the paper.
> On the other hand we have included the argument (line 196-200 in revised paper) that compared to ProDA, which consists of four structurally different training stages our model is a lot easier to  tune since it only has two loss functions.